# Demonstration of laser biospeckle method for speedy *in vivo* evaluation of plant-sound interactions with arugula

Uma Maheswari Rajagopalan[1][◐]*, Ryotaro Wakumoto[1‡], Daiki Endo[1‡], Minoru Hirai[1◐], Takahiro Kono[1‡], Hiroki Gonome[2], Hirofumi Kadono[3], Jun Yamada[1]

1 Department of Mechanical System Engineering, Shibaura Institute of Technology, Tokyo, Japan,
2 Department of Mechanical System Engineering, Yamagata University, Yamagata, Japan, 3 Graduate School of Science and Engineering, Saitama University, Saitama, Japan

◐ These authors contributed equally to this work.
‡ These authors also contributed equally to this work.
* uma@shibaura-it.ac.jp

⬲ OPEN ACCESS

**Data Availability Statement:** All relevant data are within the manuscript and its Supporting Information file.

## Abstract

In recent years, it is becoming clearer that plant growth and its yield are affected by sound with certain sounds, such as seedling of corn directing itself toward the sound source and its ability to distinguish stuttering of larvae from other sounds. However, methods investigating the effects of sound on plants either take a long time or are destructive. Here, we propose using laser biospeckle, a non-destructive and non-contact technique, to investigate the activities of an arugula plant for sounds of different frequencies, namely, 0 Hz or control, 100 Hz, 1 kHz, 10 kHz, including rock and classical music. Laser biospeckles are generated when scattered light from biological tissues interfere, and the intensities of such speckles change in time, and these changes reflect changes in the scattering structures within the biological tissue. A leaf was illuminated by light from a laser light of wavelength 635 nm, and the biospeckles were recorded as a movie by a CMOS camera for 20 sec at 15 frames per second (fps). The temporal correlation between the frames was characterized by a parameter called biospeckle activity (BA)under the exposure to different sound stimuli of classical and rock music and single-frequency sound stimuli for 1min. There was a clear difference in BA between the control and other frequencies with BA for 100 Hz being closer to control, while at higher frequencies, BA was much lower, indicating a dependence of the activity on the frequency. As BA is related to changes from both the surface as well as from the internal structures of the leaf, LSM (laser scanning microscope) observations conducted to confirm the change in the internal structure revealed more than 5% transient change in stomatal size following exposure to one minute to high frequency sound of 10kHz that reverted within ten minutes. Our results demonstrate the potential of laser biospeckle to speedily monitor in vivo response of plants to sound stimuli and thus could be a possible screening tool for selecting appropriate frequency sounds to enhance or delay the activity of plants. (337 words)

**Funding:** HK was supported by JSPS grant 19H04289. Other author(s) received no specific funding for this work.

## 1. Introduction

Environment conditions of light, wind, temperature, humidity and, CO2 concentration are essential for the efficient growth of the plant, and thus their effects are well investigated [1]. Recently, the importance of acoustic communication in plants have already been well established [2–5] and sounds are emerging as a physical trigger to improve plant health apart from the well-known chemical triggers such as plant hormones [6].

Zea mays roots exposed to frequencies of 0–900 Hz in the hydroponic system were reported to bend toward the sound with a frequency of 100–300 Hz, suggesting that sound could induce structural responses in plants [2]. Mung bean under treatments of sound with intensity around 90 dB and frequency around 2000Hz showed significant differences in stem length and root length, indicating that the audible sound wave influences the germination period of mung bean [7]. Appel and Cocroft [8] found that plants can distinguish larval chewing sounds from vibration sounds such as wind and pollen, and they showed that the leaves under threat showed an increased chemical reaction.

Sound is implicated in the activation and strengthening of the immune response in crop plants such as pepper [9], cucumber [4], and strawberry [10]. Rice plants exposed to 0.8–1.5 kHz sound waves for 1 hour showed increased tolerance to drought stress, with higher water contents and stomatal conductance than the control group [9], and sound-treated tomato showed reduced ethylene production and delayed softening compared with the control [11].

Genetical studies with Arabidopsis plants [12] reported the difference in genetical expression levels of sound-exposed and touch-treated, showing sound vibrations being a physical stimulus. Gagliano et al. [13] reported that the Pisum sativum roots could respond to environmental sound in locating water by actively growing toward flowing water below ground, implying the response of the plants to natural sound in the environment.

Sound is considered an important physical trigger that its use could be applied to delay fruit ripening instead of using chemical preservatives or genetic modification, which could have substantial economic implications. As the sound characteristics such as the frequency and the sound pressure play a significant role in achieving the desired results, there exists the necessity for determining the frequency and sound pressure levels.

Conventional methods use techniques such as measurement of dry and wet weights, number of tillers, root and shoot lengths, and ROS measurements, all of which take a considerable time for the effects to be investigated. However, in these methods, including harvesting and measuring the yield, it is necessary to carry out by dividing the growth into many growth stages, and this would take a considerable time to estimate the effects of sound stimulus and the specific characteristics of such stimulus. In addition, because of the lack of sufficient sensitivities of the techniques used, continuous real-time measurement and monitoring are almost non-existent.

To overcome such drawbacks, we propose applying laser biospeckle technique in investigating plant–sound interaction, and to our knowledge, such a study has never been done before. Due to the nature of the technique, the proposed method is non-contact and non-destructive and capable of detecting small changes within a short time.

Laser biospeckle is a dynamic interference pattern formed on the detector due to the scattered coherent light from a living body [14]. In the case of plants, there is backscattered light generated from the surface and the other generated from deep within the leaves. Both types of scattered light generate speckles, as shown in Fig 1. The intensity of such speckles changes randomly due to the movement of intracellular organelles and cellular activities such as water and nutrient transport and therefore referred to as biospeckles. By examining the dynamic changes of the biospeckle, the activity inside the plant can be monitored and has been used in

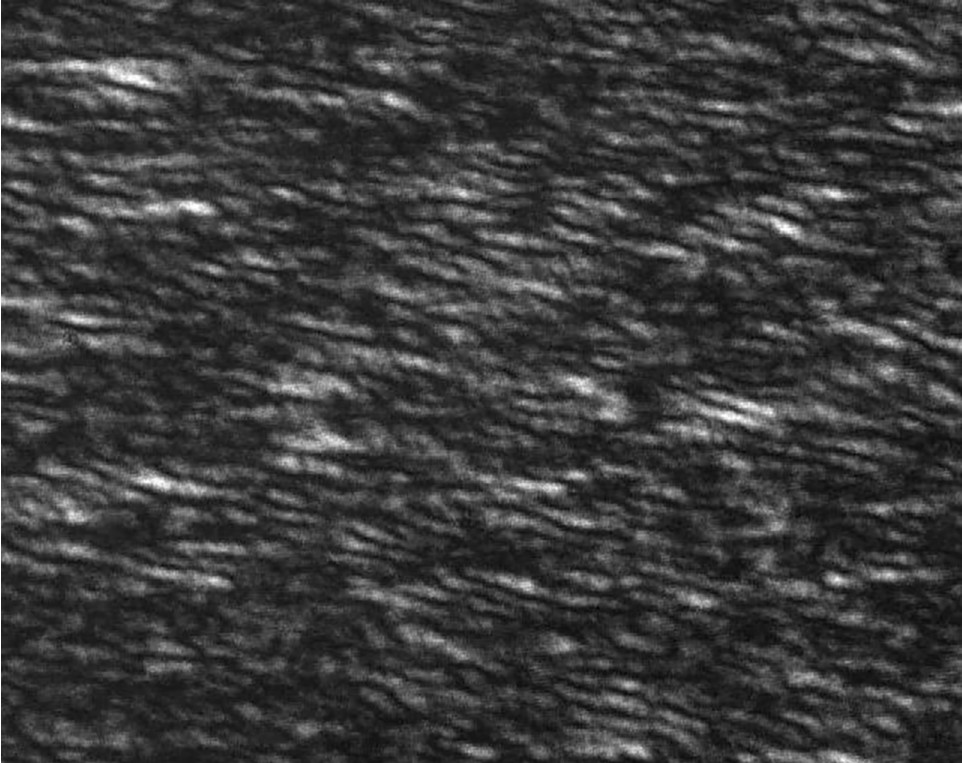

**Fig 1. A typical laser biospeckle pattern generated from plant leaves.**

agriculture include determining the quality, maturity of fruits and vegetables, analyzing seed viability, and detecting internal loss of fruits [15–17]. However, there are few studies measuring the growth and movement of intracellular organelles to study the structure of plants using the laser biospeckle method.

In this study, we propose the use of the laser biospeckle method for investigating changes within a plant leaf under sound stimulation. We investigated changes in the laser biospeckle when arugula plants were exposed to sound. Commonly known music (classical music and rock music) and single frequency (bass, middle and treble) sounds were used as stimuli. A confirmation study with a laser scanning microscope (LSM) was also conducted for comparison.

## 2. Material and method

### 2.1 Plant materials

Arugula is a plant used in French and Italian cuisine and can be cultivated relatively easily. Arugula was cultivated using hydroculture with Rockwool of size W30 × D30 × H40 (0138–008, Grotop, Grodan) from seeds. At first, to plant seeds, a cross-shaped cut or hole made by a blade across the Rockwool was placed in a plastic cup (100 ml). Arugula was grown in twelve cups (S1 Fig) and of which six were used for the experiment. Fertilizer in the form solution (OAT Agrio Co. Ltd., OAT house No.1 and No.2) was used. The nutrient solution was prepared by dissolving 1.5 g of fertilizer, OAT house No.1, and 1.0 g of OAT house No.2 fertilizer in 1.0 g of water to make the electrical conductivity measured with a water quality meter (FUSO Co., Ltd., Model-7200) to be 1.0 ds/m. Cultivation conditions of temperature, relative humidity, and photosynthetic photon flux density (PPFD) were respectively 27±5˚C, 70±5%,

and 180 mmol/(m2/s) for 24 hours. The plants were watered regularly. In all the experiments, 14 and 30 old day plants after planting seeds were used as samples.

## 2.2 Laser biospeckle mesurement

**2.2.1 System details.** Fig 2A shows a schematic of the laser speckle measurement system. A LD (laser diode) module of wavelength ($\lambda$) 635 nm (S2011-EC, Thorlabs, Japan) consisting of a focusing lens producing an elliptical beam size of 2.45 (a) mm × 0.54 mm was used as a light source. The light from the source irradiated the sample at an angle of 30˚, and the scattered light was also detected at an angle of 30˚. The scattered light was detected with a CMOS camera (DCC1545M-GL, Thorlabs, Japan, 1024 × 1280 pixel) and analyzed with a PC. The plant was mounted on a lab jockey stage with a speaker placed at a distance of around 100 mm from the plant sample under study. The distances between the sample and the laser, and that between the sample and the camera, were set as 167 mm (D). Approximate speckle size (using the relation $\lambda$ x D /a) was calculated to be 43.4μm [18].

Considering the average speckle size at the observation plane of the CMOS camera and the pixel size of the CMOS, which is around 5mm, we would like to point out that more than a sufficient number of pixels covered the speckle, thus ensuring enough signal-to-noise level. Actually, for a CMOS camera, the readout levels determine the noise level, which is 25 RMS e- per pixel for the camera used, and thus based on the pixel size and the number of pixels per speckle, the SNR of the measurements were well above the readout noise levels.

To prevent the leaf from experiencing unwanted movements during the measurement, the leaf was gently placed on a holder, as shown in Fig 2C. The holder consisted of two stainless steel plates with one plate having a window hole of diameter 2 cm and the leaf sandwiched between the plates to expose the measuring area of the leaf through the window. The hole was sufficiently large enough to accommodate the laser beam diameter. Further, to prevent reflection from the stainless-steel plates when irradiated with laser light, the stainless-steel plates were painted with a blackbody spray.

**2.2.2 Sound exposure protocol.** To study plant-sound interaction, two protocols were used and are shown respectively in Fig 3. In protocol 1, common musical sounds were used while in protocol 2 definite sound frequencies (100 Hz, 1 kHz, 10 kHz) were chosen. In either protocol, the sound pressure of 100 dB was chosen from the literature [4]. To investigate the possibility of biospeckle variation under sound exposure, the exposure timing was arbitrarily chosen as there is no prior research data available.

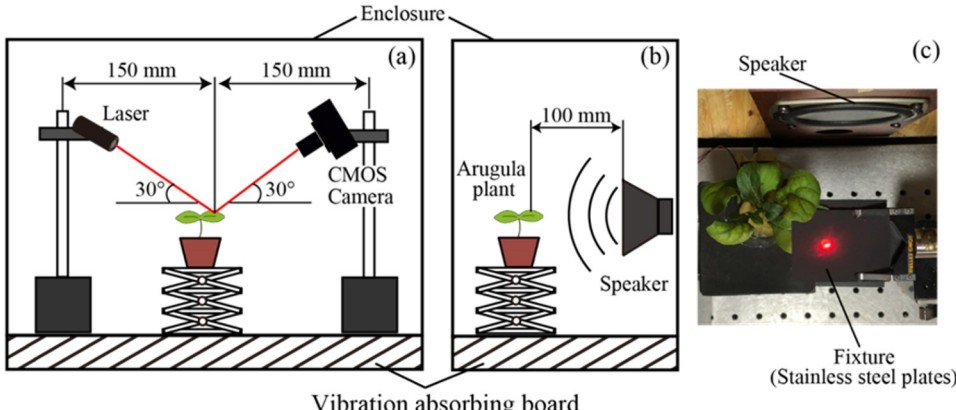

**Fig 2.** A Schematic of experimental system (a) showing the relative positions of the speaker and the arugula sample (b) with a view of the fixed arugula leaf sandwiched between two stainless plates (c).

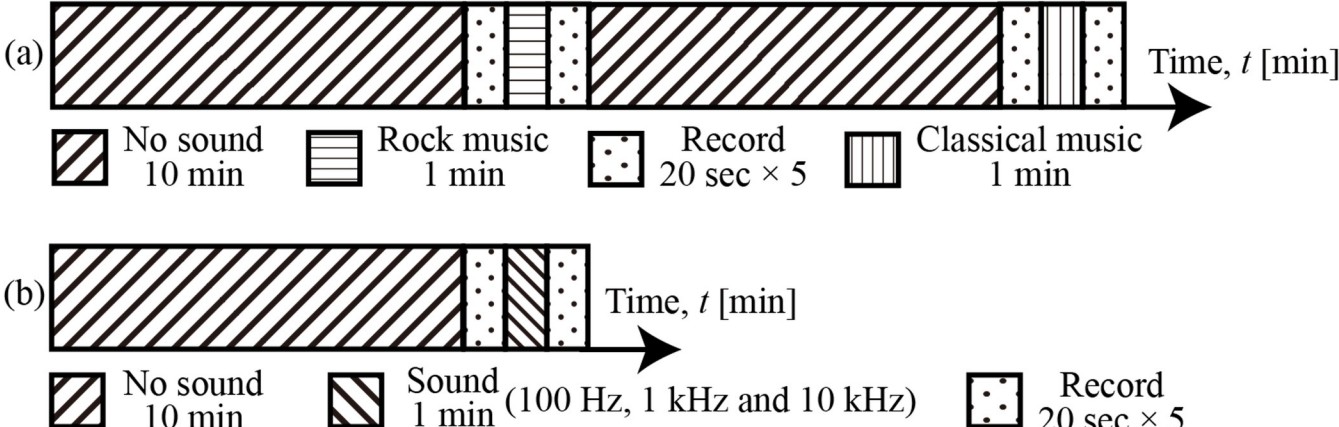

**Fig 3.** Protocol showing two cycles with each cycle starting with a blank period indicated by the slanted hatch followed by recording of biospeckles for 20 sec which is followed by a minute exposure to (a) music rock or classic and (b) frequencies 100Hz, 1KHz and 10kHz. Here both the rock and classic music and the frequencies were given sequentially.

In both protocols 1 and 2, a blank period of no sound and no light of ten minutes was used, followed by the recording of biospeckles for 20 sec at 15 fps. Exposure to sound was chosen as 1min. The acquisition was repeated five times, and the whole cycle was repeated three times. Therefore, for a single plant, for each sound, a total of 15 data were obtained. This was repeated for six replicates. A 10 min blank period was used as the interval between sound exposures to have the structural changes that might occur with exposure to sound could be restored. This blank period was also used to remove the influence of ambient noise, such as the lighting of the leaves. During the blank period, the arugula was left in a state of no illumination and no sound for 10 minutes. The arugula plants used in both the protocols were 14 and 30 days after planting (dap). Furthermore, to establish the reliability of the measurement, a diffused inanimate paper sample was used to produce speckles, and videos of speckles were recorded under the same sound exposure protocol used for plants to measure the biospeckle activity.

**2.2.3 Data analysis.** Recorded frames of the biospeckle video were analyzed by the analysis software MATLAB (R2018a (9.4.0) The MathWorks, Inc.). The correlation coefficient was calculated between the first and the rest of the subsequent frames. S2 Fig shows a schematic of the sequence of the recorded frames as a function of time with an indication of the pixel indices of a single frame.

Eq (1) shows the equation for the correlation coefficient, $r$ with $A$ being the image corresponding to the first reference frame, and $B$ indicating the image corresponding to the rest of the frames to be compared. $A_{mn}$ and $B_{mn}$ indicate the intensity of each pixel in the image and $m$ and $n$ indicate the indices. $\bar{A}$ and $\bar{B}$ correspond to the average intensity of each of the frames, namely reference and the rest of the frames.

$$r = \frac{\sum_m \sum_n (A_{mn} - \bar{A})(B_{mn} - \bar{B})}{\sqrt{\left\{\sum_m \sum_n (A_{mn} - \bar{A})^2\right\}\left\{\sum_m \sum_n (B_{mn} - \bar{B})^2\right\}}} \tag{1}$$

In this study, we introduce a quantity called Biospeckle Activity (*BA*) and is defined based on the value of correlation and is given Eq 2. When correlation is one which means that there is no movement or the object is still, then *BA* is close to zero. On the other hand, when the correlation is close to zero, indicating that there is a large deviation due to the movement of the

object under study, then *BA* becomes close to one.

$$BA = 1 - r \tag{2}$$

Here we would like to point out that the quantity BA was introduced so that it takes a positive value and thus we can discuss in terms of whether there is an increase or decrease in the internal activity of the plant, based on the increase of BA. Therefore, the parameter BA was introduced out of convenience, also prevalent in the biospeckle research field [19].

## 2.3 LSM comparison measurements

To visualize the effects of short-time exposure of one minute used in our study, LSM observations were conducted. A LSM (LEXT- OLS4000, OLYMPUS Japan). The procedure for observation of the rear side of the leaf tore off from the plant is given in Fig 4.

A total of 30 days old six seedlings were first placed in a soundproof chamber and the protocol following the timing sequence shown in Fig 4C was used to expose to sound and then observations were conducted. The plants were dark-adapted for a period of an hour followed by a stomatal size measurement of leaves every ten minutes. The other seedlings were exposed

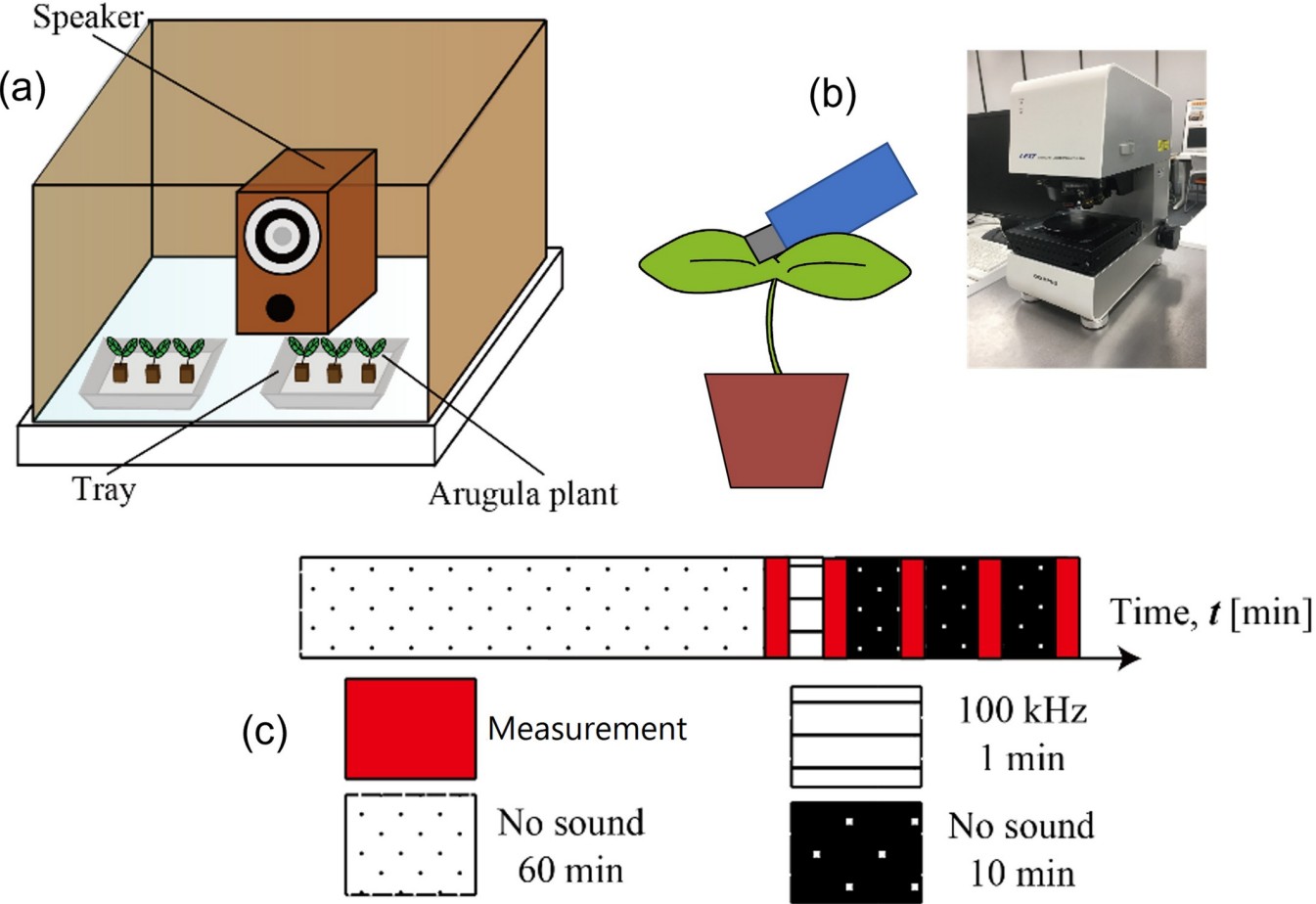

**Fig 4.** Plants placed in a soundproof chamber (a) followed by observation of a leaf teared off from the plant by OLS4000 with the sound exposure protocol shown in (c).

to a sound of frequency 10kHz and sound pressure of 100 dB for a short period of a minute. Following exposure to sound, the seedlings were kept in the dark and the stomatal sizes were measured every ten minutes.

A similar procedure was repeated except for the leaf being exposed to no sound. This would reveal the effect of exposure to laser light used in LSM observations. As more stomata are located on the rear side of leaves, observations were conducted with the rear part of a leaf. For observation of each of the seedlings, three leaves were chosen and for each leaf, sizes of five well-shaped stomata were measured.

A total of 45 stomatal sizes were obtained for each of the observation conditions. Average stomatal sizes that include the major and minor axes were calculated for control and under exposure to a sound of frequency 10kHz and sound pressure of 100 dB. The ratio of average stomatal sizes before and after exposure to one minute sound was calculated as a parameter to study the effect of sound.

## 3. Results and discussion

### 3.1 BA results under common music

To test whether the BA variation is due to the activities of the plant, speckle videos were recorded with an inanimate paper sample under the same sound protocol of all the frequencies used. The results obtained with paper are given in Fig 5. As can be seen from the BA variation of paper, the variations obtained with paper are almost close to zero, indicating no movement or indicating the stationary state of the paper. Although the variation shows a tendency of increase it was found to be within the noised level (S3 Fig). This result also establishes that the proposed measurement technique is reliable and stable.

Fig 6 shows BA as a function of time for rock music, and classical music along with control or no sound. The black solid line, blue dotted line, and green dotted line respectively indicate control, classical music, and rock music. As it can be seen from the figure, BA increases with increasing time reaching almost a plateau at around 15 seconds, with BA taking a value of around 0.9. This implies that there is increased decorrelation happening due to the movement of the organelles and other mechanisms resulting from the activity within the plant tissue.

In contrast, for the case of classical and rock music, there is a decrease of BA with the maximum values around 0.7 and 0.5, respectively, for classical and rock music. The slope of the variation of BA also differs from that of control. These changes in BA with respect to time under both classical and rock are statistically significant with respect to control under student t-test comparison of means with p value less than 0.05.

Since classical music and rock music contain various frequencies and sound pressure levels, it is unclear what kind of sound affects BA. Therefore, we investigated BA under three different frequencies low, medium, and high frequency. Sound pressure was kept the same. BA was measured when the arugula plant was exposed to those frequencies.

### 3.2 BA results under different frequencies

Fig 7A–7D shows the variation of BA as a function of time for four types, namely, control or no sound, 100 Hz, 1 kHz, and 10 kHz for arugula plants 14th and 30th days old after planting the seed shown respectively in red and green lines. From Fig 7A and 7D, it can be seen that BA changes depending on the growth stage. In other words, the activity in response to sound within the leaf changes with age. The variation in BA with respect to time follows the same trend as shown in Fig 6 as that show for rock and classical sounds. Also, from figures (a) and (d) of Fig 7, BAs of leaves exposed to sound are lower than those not exposed to sounds.

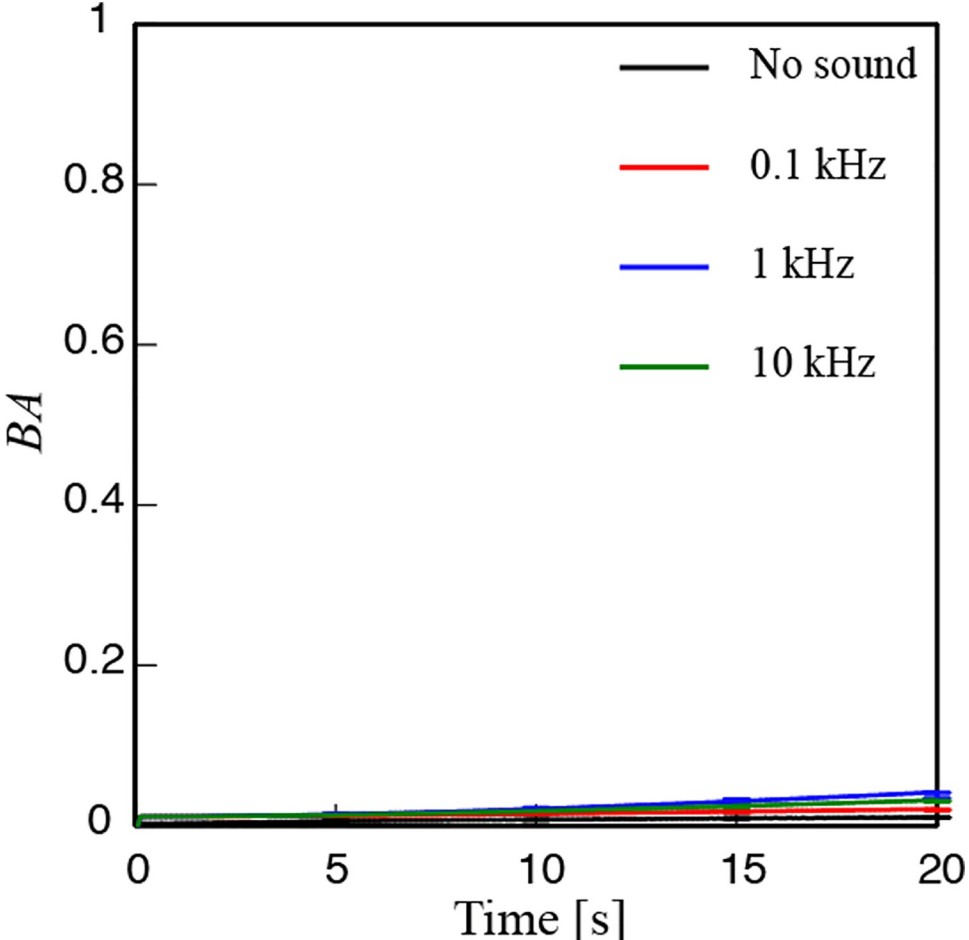

**Fig 5. Temporal variation of BA from a phantom object of paper under exposure to different frequency sounds used in the experiment.** Error bars correspond to standard error of the mean.

In addition, depending on the frequency and the age of the plant, the variation of BA is different. At the low frequency of 0.1kHz, the variation shows a linear increase with increasing time reaching a plateau at around 15 seconds under both 14 and 30 dap. Here it should be mentioned that the ambient sound level was around 43dB. The variation 0.1kHz was almost close to that of under no sound for 14 dap while that for 30dap, the effects are different, indicating the dependency of the plant age.

Similarly, both for the middle and high frequencies, the behavior is almost the same with monotonous variation at the beginning followed by a constant one in the latter part of the time of observation, although the higher frequency has a larger effect on the slope of the variation taking longer to reach the plateau. Here again, the age effects of the plants are reflected in the BA activities, with the high frequency having a more significant effect on an older plant of 30dap.

Our results demonstrate that the variation of BA does depend on the age of the plant. Therefore, at all the frequencies used, 100 Hz, 1 kHz, and 10 kHz, the effect of sound was strongly seen with monotonously increasing BA in the first 5 seconds and beyond five seconds, BA tending to reach a plateau. A t-test was used to investigate the significance of BA under differences for young and old leaf against control or no sound condition, and the results are shown in S1 Table.

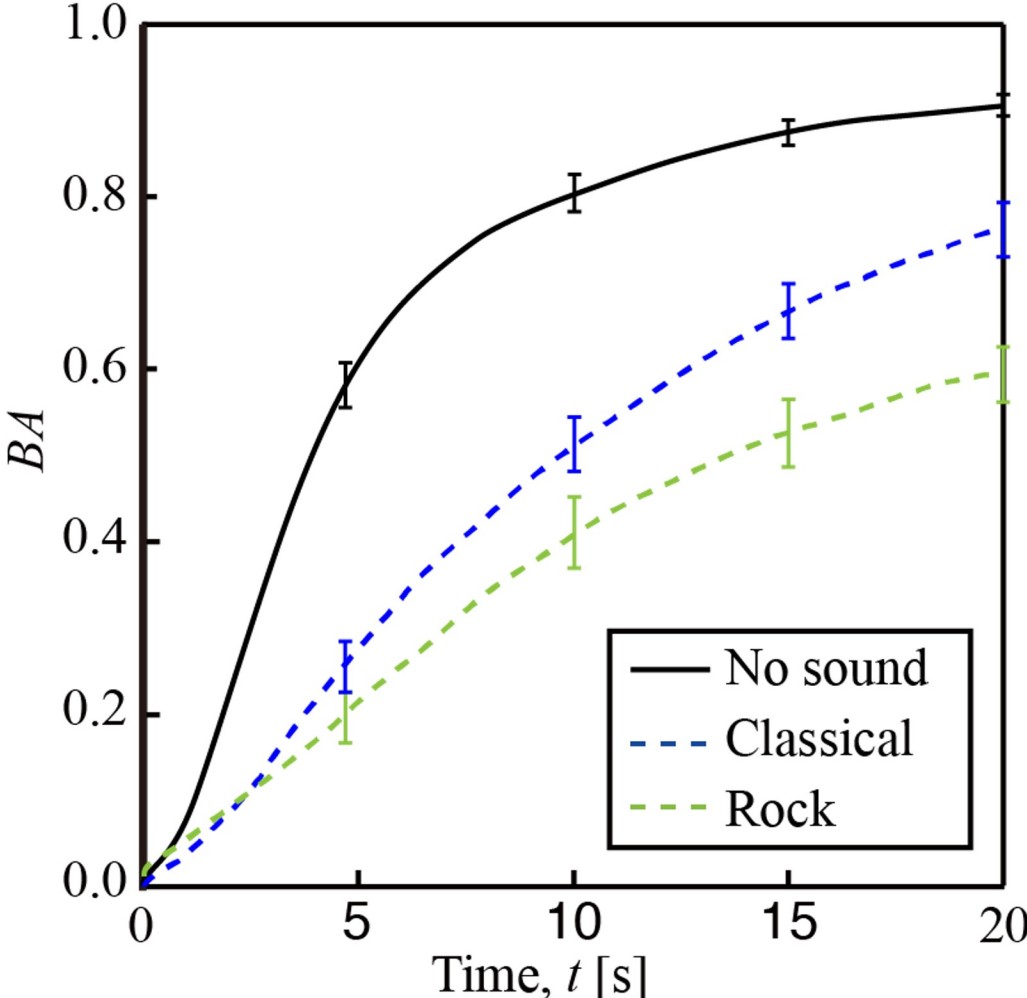

**Fig 6. Temporal variation of BA from leaf of an arugula plant under exposure to sounds of classical and rock music.**
Black solid line, blue dotted line, and green dotted line respectively indicate control, classical music, and rock music.
Error bars correspond to standard error of the mean.

As could be seen, the variation of BA was almost significant except for the case under 100kHz. Next, variation of BA within five seconds was considered for comparing the effect of different frequencies. To investigate the influence of each frequency, normalization was performed. Normalization was done by the value of BA under a particular frequency by that of the BA under control or no sound.

Fig 8 shows the normalized BA as a function of frequency for two different ages of the plants, 14 dap and 30 dap. The red square and green circle respectively show arugula of 14 and 30 dap. There is a difference in the dependency of the normalized BA depending on the age of the plant and also on the frequency. For 14 dap, there is a slight decrease in the normalized BA with increasing frequency and these differences were found to be not statistically significant. On the other hand, for 30 dap, there is a gradual decrease in the ratio with increasing frequency with the decrease being the largest for 10kHz which was found to be statistically significant (t-test, $p < 0.5$). Therefore, higher frequency has a significant effect on the biospeckle activity of the plant leaf and thus, the frequency dependence of BA showed a clear age dependence with higher BA, for the younger plant. As we could see the largest change in BA with

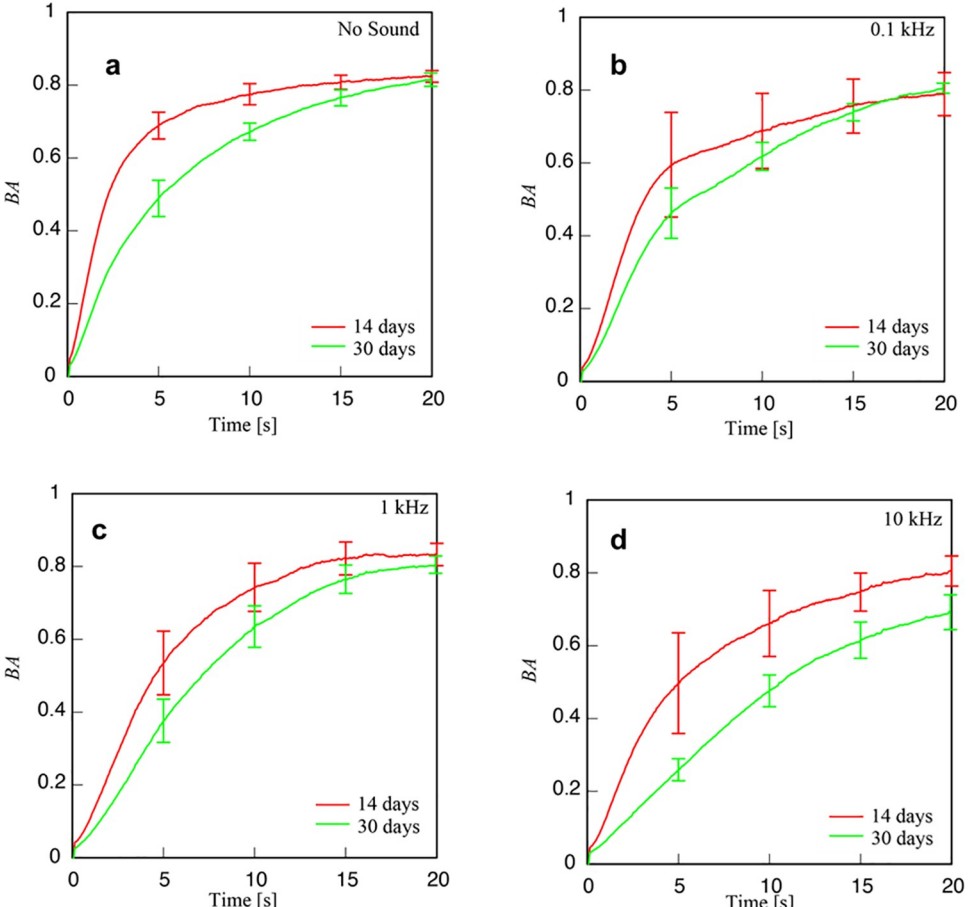

**Fig 7.** Temporal variation of BA from leaf of an arugula plant under exposure to sounds of different frequencies with (a) control or no sound, (b) 0.1kHz, (c) 1 kHz, and (d) 10 kHz for 14 dap (red) and 30 dap (green).

10kHz, LSM observations were performed for comparison. It should be pointed out, that while our biopspeckle method could be conducted noninvasively in vivo, LSM requires plucking the leaf from the plant and conducting observations.

### 3.3 LSM observations and comparison

Fig 9 shows a typical image obtained by the LSM. Several elliptically structured stomata as indicated by red arrowheads can be visualized. For each of the five well-shaped stomata, the major and minor axes of the ellipse were measured for the case of just before exposure to sound and after exposure to a minute exposure of 10kHz.

Fig 10 A shows the major and minor axes lengths represented respectively in red square and green circle as a function of time with the top left for 10kHz sound and bottom left for control. The gray shaded region corresponds to the exposure period. Here the sizes were sampled every ten minutes after exposure up to a period of 30 min. Comparing the results under control and those at 10kHz sound exposure, a clear decrease in the sizes of both the major and minor axes could be seen. Further it can be seen that under control, over the whole observation period of 30 min, no change in the sizes could be seen.

In order to make the difference cleared due to exposure, a ratio was calculated by dividing the stomatal sizes after by those before exposure to sound (Fig 10B). If there is no change due

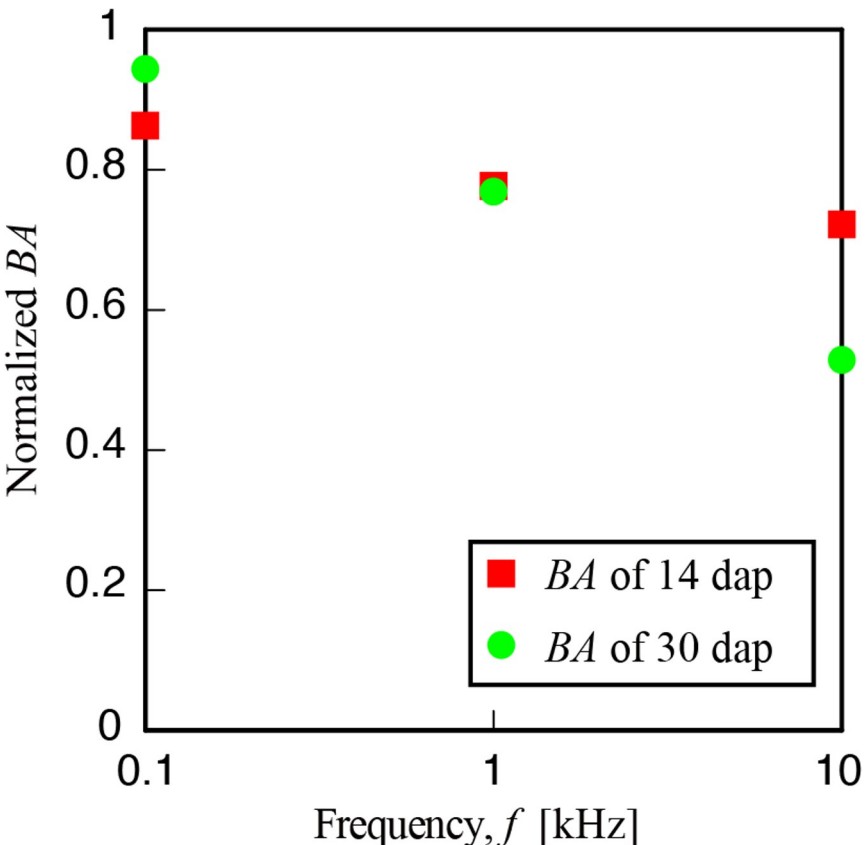

**Fig 8. Normalized BA as a function of frequency for arugula plant at 14 days (red square) and 30 days (green circle).** Only the result for 10kHz is statistically significant.

to sound, the ratio should be close to one. However, as we can see the ratio fluctuates around one sometimes becoming larger than one. This indicates that the returning of the stomatal sizes to the values close to before exposure is not happening monotonously but more as of expansions and contractions. In other words the stomata tended to return to the original sizes more in a fluctuating manner. For 10kHz, a reduction of more than 5% can be seen. A t-test comparison revealed the difference to be statistically significant (p<0.5). Moreover, with time, those that were exposed to 10kHz, the stomatal sizes reverted to close to control value or in other words pre-exposure levels almost within a period of around 10 min.

Stomatal observations indicated that even one-minute exposure to sound can produce significant changes in the anatomical structure of the leaf. Such stomatal changes have been observed with relatively longer exposure durations [20]. Our observations revealed that the stomatal size changes could be dynamic with the changes being reversible after a short exposure to sound stimuli. Because the plants were first dark-adapted, it is possible that the stomatal closure could be due to sudden exposure to the laser light used for observation itself. However, conducting under similar conditions under no sound condition did indeed confirm the effects were due to sound used as stimulus.

As biospeckles were found to show the largest effect for a sound frequency of 10kHz, the stomatal observations were conducted with 10kHz. Such observations indeed confirmed even a short time exposure of one min was enough to cause structural changes in the leaf. Speckles are produced because of scattering from different structures within the leaf and any dynamic

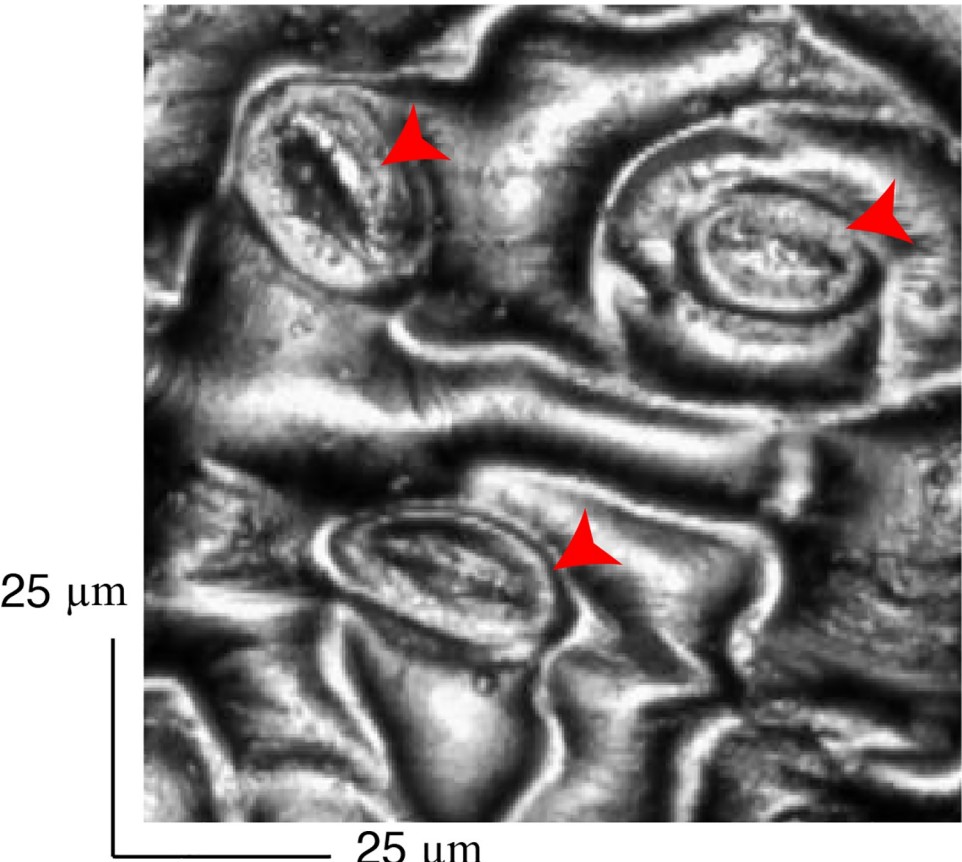

**Fig 9. A typical observation of the rear part of the leaf with the laser scanning microscope.** The red arrowheads indicate the elliptically shaped stomata.

structural change would result in a change of speckle intensity. This in turn would result in the reduction of correlation between the temporally recorded speckle patterns.

Although the current study was conducted for a short duration of one min of sound exposure, we expect laser biospeckles to show sufficient changes for longer periods as well as day-to-day changes for the case of prolonged study. We would also like to point out the difference in the dark period used before measurement. A sufficient period of 1 hour was used for adapting to dark in the case LSM while for the laser speckle measurement, 10 min was found to be sufficient enough to have an effect on plant response because of the superior sensitivity of biospeckles in detecting changes with sound exposure. Also, our LSM observations indicate a restoration of changes within ten minutes.

The advantage of short-time exposure could be used as a screening tool for making the appropriate frequency that can be used either to enhance or suppress the plant activities. For example, it may be possible to have a frequency that would enhance the suppression of weed growth while at the same time, a different frequency could be employed to facilitate the growth behavior of the plant of interest.

## 4. Conclusion

In this study, we proposed using laser biospeckle to investigate the response of plants to sound stimuli. Laser biospeckles were recorded from plant leaves at a wavelength of 635 nm when the

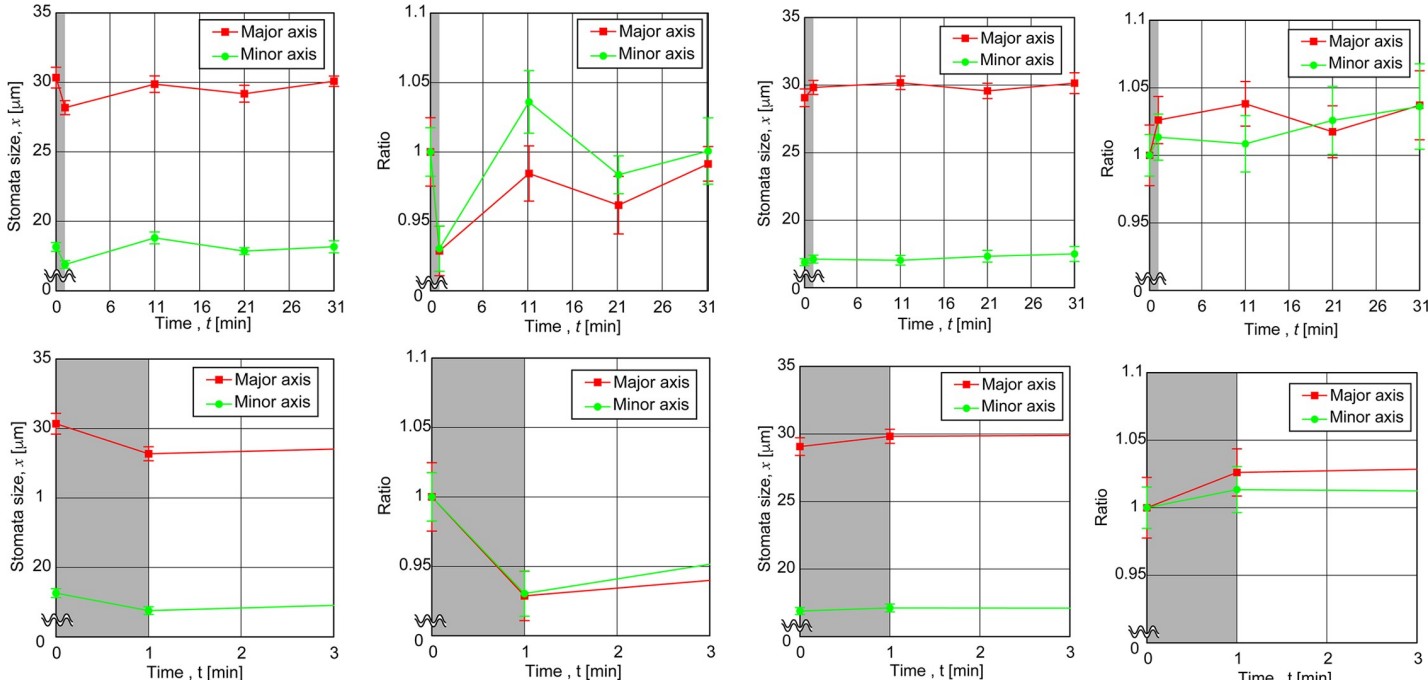

**Fig 10.** The stomatal sizes, major and minor axes lengths represented respectively in red square and circle as a function of time shown on the top left and the ratio of stomatal lengths on the top right for 10kHz sound (A) and for control (B). The gray shaded regions shown at the top for the stomatal lengths and ratio are magnified to show respectively the first three seconds in the bottom row. Here the ratio is taken with the stomatal sizes measured after exposure to those before exposure.

plant was exposed to different sound stimuli of classical and rock music and single-frequency sound stimuli for a minute and the biospeckle activity (*BA*) was calculated. It was found that the *BA* values were lower than that of the control when the sound was exposed, suggesting a reduction in the activity of the plant tissue. In addition, it was shown that there were changes in the activities of the plant tissue at each frequency (0.1 kHz,1 kHz and 10 kHz) and that such dependence varied depending on the age of the plant. These results indicate that laser biospeckle method could be a potential tool for speedily examining the in vivo response of plants to sound stimuli. Further, LSM observations conducted to confirm the change in the internal structure revealed around five percent change in the stomatal size with one minute of 10 kHz sound exposure. This could be one possible origin for the changes in *BA* signals apart from other physiological changes, for example, flow of fluids and cytoplasmic streaming, which remains unknown.

## Supporting information

**S1 Fig. Tray containing twelve cups with planted seeds in rockwool.** Three seeds were sown per cup.
(TIF)

**S2 Fig.** A sequence of recorded frames as a function of time (left) with the pixel indices of each frame indicated as (m,n).
(TIF)

**S3 Fig. The noise level of the dark pixel of the camera with superimposed BA of a paper.**
(TIF)

**S1 Table. p-values of the results of t-test done under different frequencies with that under control of no sound for two different leaves of different ages.**
(DOCX)

## Author Contributions

**Conceptualization:** Uma Maheswari Rajagopalan, Jun Yamada.

**Data curation:** Uma Maheswari Rajagopalan, Ryotaro Wakumoto, Daiki Endo, Minoru Hirai.

**Formal analysis:** Uma Maheswari Rajagopalan, Daiki Endo, Minoru Hirai.

**Funding acquisition:** Hirofumi Kadono, Jun Yamada.

**Investigation:** Uma Maheswari Rajagopalan, Ryotaro Wakumoto.

**Methodology:** Uma Maheswari Rajagopalan.

**Project administration:** Uma Maheswari Rajagopalan.

**Software:** Uma Maheswari Rajagopalan.

**Supervision:** Uma Maheswari Rajagopalan, Takahiro Kono, Jun Yamada.

**Validation:** Uma Maheswari Rajagopalan.

**Visualization:** Uma Maheswari Rajagopalan, Ryotaro Wakumoto, Daiki Endo, Minoru Hirai.

**Writing – original draft:** Uma Maheswari Rajagopalan.

**Writing – review & editing:** Takahiro Kono, Hiroki Gonome, Hirofumi Kadono, Jun Yamada.

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
