## [Decision Letter · Decision Letter 0]

12 May 2021

PONE-D-21-11287

Demonstration of Laser biospeckle method for speedy in vivo evaluation of plant-sound interactions with argula

PLOS ONE

Dear Dr. RAJAGOPALAN,

Thank you for submitting your manuscript to PLOS ONE. After careful consideration, we feel that it has merit but does not fully meet PLOS ONE’s publication criteria as it currently stands. Therefore, we invite you to submit a revised version of the manuscript that addresses the points raised during the review process.

We look forward to receiving your revised manuscript.

Kind regards,

Xuejian Wu, Ph.D.

Academic Editor

PLOS ONE

Journal Requirements:

3. From your data statement, it is unclear why you have declared 'No - some restrictions will apply' . Please clarify the nature of these restrictions, ie. If due to ethical or legal reasons.

PLOS defines a study's minimal data set as the underlying data used to reach the conclusions drawn in the manuscript and any additional data required to replicate the reported study findings in their entirety. All PLOS journals require that the minimal data set be made fully available. For more information about our data policy, please see http://journals.plos.org/plosone/s/data-availability.

5. Please include captions for your Supporting Information files at the end of your manuscript, and update any in-text citations to match accordingly. Please see our Supporting Information guidelines for more information: http://journals.plos.org/plosone/s/supporting-information

Reviewers' comments:

Reviewer's Responses to Questions

**Comments to the Author**

1. Is the manuscript technically sound, and do the data support the conclusions?

Reviewer #1: Yes

Reviewer #2: Partly

2. Has the statistical analysis been performed appropriately and rigorously? 

Reviewer #1: No

Reviewer #2: No

3. Have the authors made all data underlying the findings in their manuscript fully available?

Reviewer #1: Yes

Reviewer #2: Yes

4. Is the manuscript presented in an intelligible fashion and written in standard English?

Reviewer #1: Yes

Reviewer #2: No

5. Review Comments to the Author

Reviewer #1: The paper is good, but before publication I suggest authors to give more details about the CMOS used, including focal length of the lens, and the estimation of the speckle size produced by the imaging system. Finally, I suggest authors to consider more statistical analysis on the data to make more reliable their results.

Reviewer #2: Speckle imaging has been proved to be an effective way for structure measurement. In biology, it has been used to distinguish different cellular movement patterns and indicate dynamic processes. In this manuscript, authors used speckle imaging to study the interaction between plant leaves and various acoustic frequencies. They show that speckle imaging can perform real-time monitoring.

After reading this manuscript, several questions arise. I hope they can be answered appropriately.

1. Specifications of the speckle imaging system are required, including accuracy, stability, sensitivity, etc. For instance, Figure 6 claims the difference between leaves at different stages. However, it is possible that the authors merely measured the shot noise limit of the CMOS camera because all curves tend to overlap at BA=0.8. Different rising edges can be caused by different scattering ratio which may not relate to structural difference.

2. Phantom tests are also required to prove the performance of the system. Simple specimens, such as gel or metal, can be used to verify system performance before the biological tests. For instance, even though absorption boards are used to reduce vibration, system instability can also lead to a smaller BA when the stimuli frequency is 10 kHz. If the system can show a constant BA regardless of the stimuli frequencies when conducting phantom tests, BA values under different stimuli will be more trustworthy.

3. The reason for selecting 0 Hz, 100 Hz, 1 kHz and 10 kHz needs to be stated. As shown in the Introduction, meaningful phenomena were discovered within hundreds of Hz. Instead of using high frequencies, a high frequency resolution and a range within hundreds of Hz are preferred.

4. What is the necessity of creating BA? According to Eq. (2), it simply equals to 1-r, and r is the correlation coefficient.

5. The full name of an aberration needs to be listed when the aberration occurs first. In Line 42 and Line 95, there are missing full spells.

6. Grammar errors. The Abstract itself contains many grammar errors. Several errors in the Abstract are listed in the following. Line 25, it should be “are destructive” instead of “destructive”. Line 30, it should be “intensities of speckles change over time” instead of “intensity of speckles change in time”. Line 34, there is a missing article. Line 42, “a leaf” instead of “leaf” should be used.

6. PLOS authors have the option to publish the peer review history of their article (what does this mean?). If published, this will include your full peer review and any attached files.

Reviewer #1: No

Reviewer #2: No

---

## [Author Response · Author response to Decision Letter 0]

11 Sep 2021

Please find the attached response

---

## [Decision Letter · Decision Letter 1]

27 Sep 2021

PONE-D-21-11287R1Demonstration of Laser biospeckle method for speedy in vivo evaluation of plant-sound interactions with argulaPLOS ONE

Dear Dr. RAJAGOPALAN,

Thank you for submitting your manuscript to PLOS ONE. After careful consideration, we feel that it has merit but does not fully meet PLOS ONE’s publication criteria as it currently stands. Therefore, we invite you to submit a revised version of the manuscript that addresses the points raised during the review process.

We look forward to receiving your revised manuscript.

Kind regards,

Xuejian Wu, Ph.D.

Academic Editor

PLOS ONE

Journal Requirements:

Additional Editor Comments (if provided):

Reviewers' comments:

Reviewer's Responses to Questions

**Comments to the Author**

1. If the authors have adequately addressed your comments raised in a previous round of review and you feel that this manuscript is now acceptable for publication, you may indicate that here to bypass the “Comments to the Author” section, enter your conflict of interest statement in the “Confidential to Editor” section, and submit your "Accept" recommendation.

Reviewer #1: All comments have been addressed

Reviewer #2: (No Response)

2. Is the manuscript technically sound, and do the data support the conclusions?

Reviewer #1: Yes

Reviewer #2: Yes

3. Has the statistical analysis been performed appropriately and rigorously? 

Reviewer #1: Yes

Reviewer #2: Yes

4. Have the authors made all data underlying the findings in their manuscript fully available?

Reviewer #1: Yes

Reviewer #2: Yes

5. Is the manuscript presented in an intelligible fashion and written in standard English?

Reviewer #1: Yes

Reviewer #2: Yes

6. Review Comments to the Author

Reviewer #1: The authors addressed all the comments adequately. The paper is good, well organized, the results are interesting and novel and I suggest acceptance of the paper

Reviewer #2: The authors reply my concerns properly with newly-added details and improved English writing. Compared with the former version, the latest one is more convincible. However, there are still three questions I would like the authors to expand on.

1. For the phantom test shown in Fig. 5, it looks like that the BA increases a little along with the testing time. Is it necessary to subtract the phantom results from the Arugula results to correct the final BA values? Moreover, what are the reasons for the increase in the phantom test?

2. For the LSM results shown in Fig. 10, the control group shows ratios larger than 1. As there were no sound stimuli, what are the reasons for the ratios larger than 1?

3. The gray regions in Fig. 10 are too narrow to see. The resolutions of all the figures need to be improved.

7. PLOS authors have the option to publish the peer review history of their article (what does this mean?). If published, this will include your full peer review and any attached files.

Reviewer #1: No

Reviewer #2: No

---

## [Author Response · Author response to Decision Letter 1]

6 Oct 2021

Please find the attached file ReplyToReviewer2commetns

---

## [Editor Report · Decision Letter 2]

11 Oct 2021

Demonstration of Laser biospeckle method for speedy in vivo evaluation of plant-sound interactions with argula

PONE-D-21-11287R2

Dear Dr. RAJAGOPALAN,

We’re pleased to inform you that your manuscript has been judged scientifically suitable for publication and will be formally accepted for publication once it meets all outstanding technical requirements.

Kind regards,

Xuejian Wu, Ph.D.

Academic Editor

PLOS ONE
---

## [Editor Report · Acceptance letter]

19 Oct 2021

PONE-D-21-11287R2 

Demonstration of Laser Biospeckle Method for speedy in vivo evaluation of plant-sound interactions with argula 

Dear Dr. Rajagopalan:

I'm pleased to inform you that your manuscript has been deemed suitable for publication in PLOS ONE. Congratulations! Your manuscript is now with our production department. 

Kind regards, 

on behalf of

Dr. Xuejian Wu 

Academic Editor

PLOS ONE